# The Growth of Extended Melem Units on g-C_3_N_4_ by Hydrothermal Treatment and Its Effect on Photocatalytic Activity of g-C_3_N_4_ for Photodegradation of Tetracycline Hydrochloride under Visible Light Irradiation

**DOI:** 10.3390/nano12172945

**Published:** 2022-08-26

**Authors:** Thi Van Anh Hoang, Phuong Anh Nguyen, Won Mook Choi, Eun Woo Shin

**Affiliations:** School of Chemical Engineering, University of Ulsan, Daehakro 93, Nam-gu, Ulsan 44610, Korea

**Keywords:** tri-s-triazine unit, hydrothermal treatment, tetracycline hydrochloride, visible light photocatalysis, charge carrier

## Abstract

In this work, the growth of extended tri-s-triazine units (melem units) on g-C_3_N_4_ (CN) by hydrothermal treatment and its effect on the photodegradation efficiency of tetracycline hydrochloride (TC) is investigated. The CN-180-x and CN-200-6 samples were prepared using different hydrolysis times and temperatures, and they were characterized by multiple physicochemical techniques. In addition, their photodegradation performance was evaluated under visible light irradiation. Compared to the CN, CN-180-6 possesses remarkable photocatalytic degradation efficiency at 97.17% towards TC removal in an aqueous solution. The high visible-light-induced photo-reactivity of CN-180-6 directly correlates to charge transfer efficiency, numerous structural defects with a high specific surface area (75.0 m^2^ g^−1^), and sufficient O-functional groups over g-C_3_N_4_. However, hydrothermal treatment at a higher temperature or during a longer time additionally induces the growth of extended melem units on the surface of g-C_3_N_4_, resulting in the inhibition of the charge transfer. In addition, the superoxide radical is proven to be generated from photoexcited reaction and plays a key role in the TC degradation.

## 1. Introduction

The presence of antibiotic residues in the environment has received widespread attention due to the undesirable effects on human health and livestock security [1,2,3,4]. Tetracycline hydrochloride (TC) is one of the most common pharmaceutical residues, which is in wide usage on a global scale [5]. Therefore, a variety of methods have been developed to remove TC from an aqueous environment, including membrane separation, biodegradation, adsorption, Fenton oxidation, and photocatalyst degradation [6,7,8,9,10]. Among them, the photodegradation process is considered as a superior strategy for elimination of TC based on its low cost, sustainable technology, and high removal efficiency [11,12].

The well-known metal-free semiconductor graphitic carbon nitride (g-C_3_N_4_) has been demonstrated as a promising material for photocatalysis due to its suitable band gap (2.7 eV) for visible light, high physicochemical stability, and ecological friendliness. Nevertheless, the photocatalytic ability of g-C_3_N_4_ is restricted in terms of the high recombination of charge carriers, low specific surface area, and inadequate usage of visible light [13,14]. To overcome these issues, numerous attempts have been made to facilitate the photocatalytic behavior of g-C_3_N_4_ via heterostructure construction, defects engineering, and doping technology [15,16,17]. In terms of the enhancement of the overall performance of materials, the hydrothermal process has been introduced as an effective and uncomplicated method to significantly improve the photocatalytic properties of the g-C_3_N_4_ photocatalyst. This facile and green method can etch numerous defects on the g-C_3_N_4_ surface and introduce desirable O-containing functional groups to improve photodegradation efficiency [13,18]. However, it was reported that strong hydrolysis in the method also degraded a regular array of tri-s-triazine (melem) units in the g-C_3_N_4_ structure [17].

In this study, the extension of the melem unit and further growth of the extended melem unit during the hydrothermal treatment and the effect on the photocatalytic behaviors were carefully evaluated for the photodegradation of TC in an aqueous solution under visible light irradiation. Various hydrothermal treatment times for modifying the g-C_3_N_4_ were chosen at 3, 6, and 9 h. The photodegradation results showed that the g-C_3_N_4_ treated hydrothermally for 6 h was the best for TC removal efficiency, resulting from changes in the structure and properties of g-C_3_N_4_. The high specific surface area caused by the generation of porous defects and the introduction of O-functional groups on the surface of g-C_3_N_4_ gave a positive effect on the photodegradation performance. In contrast, the appearance of the extended melem unit and its further growth over g-C_3_N_4_ during the hydrothermal treatment, unfortunately, had a negative influence on the charge transfer efficiency over the g-C_3_N_4_, resulting in a high recombination rate of the photogenerated electron–hole pairs.

## 2. Materials and Methods

### 2.1. Synthesis of CN-180-x and CN-200-6

Pristine g-C_3_N_4_ was synthesized by a thermal polycondensation method. The precursor, thiourea (30 g) (CH_4_N_2_S, ≥99%, supplied by Sigma-Aldrich Korea, South Korea), was placed in a crucible covered with aluminum foil and heated to 550 °C for 4 h (ramping rate = 5 °C/min) in a muffle furnace, using air as the atmosphere. The as-synthesized bulk g-C_3_N_4_ sample was named CN.

One gram of CN powder was dissolved in 100 mL of DI water and then sonicated for 2 h at room temperature. The 150 mL Teflon^™^ autoclave containing the CN-180-x samples (x = 3, 6, or 9 h where x was the hydrothermal period) was heated in an oven to 180 °C (ramping rate = 5 °C/min). In comparison with the CN-180-6, the CN-200-6 was synthesized in 6 h at 200 °C. The resulting materials were repeatedly rinsed with DI water after cooling to room temperature before being freeze-dried for the photocatalytic activity test and further characterization. The schematic procedure for CN and catalysts synthesis was illustrated Figure 1. 

### 2.2. Characterization

A QUADRASORB^™^ SI surface area instrument (Quantachrome Instruments, Boynton Beach, FL, USA) and N_2_ adsorption-desorption isotherms were applied to measure the Brunauer–Emmett–Teller (BET) specific surface areas. The morphologic changes of the CN-180-x and CN-200-6 were determined by transmission electron microscopy (TEM) Tecnai G2 F20 X-TWIN (FEI, Austin, TX, USA). A Cu K X-ray source with a wavelength of 1.5415 (scan rate of 2° (2θ)/min) was used to study the X-ray diffraction (XRD) patterns of the photocatalytic samples using a D/MAZX 2500 V/PC high-power diffractometer from Rigaku (Tokyo, Japan). A Fourier transform infrared (FTIR) transmittance spectrometer (Nicolet^™^ 380 spectrometer, Nicolet^™^ iS5 with an iD1 transmission accessory by Thermo Scientific^™^, Waltham, MA, USA) was employed to determine the functional groups of the produced photocatalysts. X-ray photoelectron spectroscopy (XPS) was carried out using Thermo Scientific’s K-Alpha system. 

A UV–Vis diffuse reflectance spectrometer (UV-DRS) (SPECORD^®^ 210 Plus spectroscope; Analytik Jena, Germany) examined the optical characteristics of the photocatalysts. The electrochemical impedance spectroscopy (EIS) data was conducted on an impedance analyzer (VSP series; Bio-Logic Science Instruments, Seyssinet-Pariset, France). A fine powder mixture containing 20 mg catalyst sample and 2 mg of active carbon was added into 100 µL isopropanol 99.7% and 30 μL Nafion 5 wt% (both from Sigma-Aldrich Korea, Gyeonggi, South Korea). The electrolyte for the three-electrode electrochemical cell consisted of 10 mL of a 1 M NaOH solution. While a RE-1BP (Ag/AgCl) electrode was the reference electrode, the platinum wire was the counter electrode, and the working electrode was a 6 mm standard glassy carbon electrode. To measure the EIS, 10 μL of the obtained mixture was carefully coated onto the glassy carbon electrode. Then, setting a frequency range of 0.01 Hz to 100 kHz with a 10 mV amplitude and a direct current potential of +0.8 VSCE was utilized. For additional verification of the charge carrier recombination rate, time-resolved photoluminescence (PL) spectra were analyzed by an FS5 spectrofluorometer (Edinburgh Instruments Ltd., Livingston, UK) under 400 nm laser excitation and the emission decay was fitted with tri-exponential functions. 

### 2.3. TC Photodegradation Test

The photocatalytic activities were conducted by an Oriel’s Sol1A™ Class ABB system with a 140 W xenon lamp and UV cut-off filter (λ > 420 nm) as the light source. In a typical photodegradation, 10 mg of as-prepared samples were suspended in 50 mL of 20 ppm TC. Before the photocatalytic reaction, the suspension was stirred for 60 min in the dark phase to achieve adsorption–desorption equilibrium and then was irradiated for 2.5 h at the ambient temperature. At 30 min intervals, 3 mL of the solution was filtered through a polytetrafluoroethylene membrane filter (Whatman GmbH, Dassel, Germany) and analyzed by absorbance at λ_max_ = 357 nm as a function of the amount of radiation it received, which was measured using a SPECORD 210 Plus spectroscope. The pseudo-first-order kinetic model was followed by the Equation (1):(1)$$\ln\frac{C}{C_{0}}=-kt$$
where *C*_0_ (mg/L) was the initial concentration of TC before irradiation, *C* was the actual TC concentration at time *t*, and *k* was the kinetic rate constant (min^−1^) [19,20].

In addition, to confirm the reusability of the as-prepared samples, a 4-cycle experiment was carried out for 50 min for each run under the same reaction conditions.

### 2.4. Reactive Species Test

The role of the reactive species in the removal of TC over the CN-180-6 samples was determined by some specific scavengers. Specifically, 5 mM Teoa triethanolamine (TEOA) (h+ quenching agent), 5 mM isopropyl alcohol (IPA) (•OH quenching agent), and 5 mM p-benzoquinone (BQ) (•O_2_^−^ quenching agent) were dropped into the solution for the quenching experiments. 

## 3. Results and Discussion

### 3.1. Physicochemical Properties

The N_2_ adsorption–desorption isotherms of the as-prepared photocatalysts were depicted in Appendix A. All the samples obviously exposed type IV isotherms with a hysteresis curve of the type H3 pattern, confirming the mesoporous characteristic of the catalysts. Otherwise, the specific BET surface area (S_BET_), pore volume (V), and average pore diameter (L) of the samples are represented in Table 1. During the hydrothermal treatment, the S_BET_ and V values for the CN-180-x and CN-200-6 samples had a volcano pattern with a maximum at CN-180-6. The highest values were 75.0 m^2^ g^−1^ of S_BET_ and 0.60 cm^3^ g^−1^ of V, respectively, which were almost 3 times higher than those for CN (28.5 m^2^ g^−1^ and 0.20 cm^3^ g^−1^). The high specific surface area can enhance the light-harvesting and improve the photodegradation performance of the photocatalyst. However, the CN-180-9 and CN-200-6 showed lower S_BET_ and V values due to a longer hydrolysis time and a higher hydrothermal temperature. This could be related to the strong hydrolysis/oxidation degree and the attachment of O-containing groups over g-C_3_N_4_, which will be clarified later in the TEM images and XPS data.

The morphological structures of the CN-180-x and CN-200-6 samples were examined by TEM images. As displayed in Figure 2, all of the as-prepared samples exhibited a typical two-dimensional sheet-like structure. In comparison to the clean surface of the CN-180-3 (Figure 2a), the CN-180-9 and CN-200-6 showed the obvious pothole structure on the surface of g-C_3_N_4_ due to the longer hydrothermal time and the higher temperature, resulting from further hydrolysis on the g-C_3_N_4_ surface structure (Figure 2c,d). However, the CN-180-6 displayed less pothole structure but significant unevenly porous cracks on the surface at the high magnification, implying the highest specific surface area for CN-180-6. 

The crystal structures of all synthesized photocatalysts were analyzed by the XRD measurement (Figure 3A). For pristine CN, two characteristic peaks appearing at 12.6 and 27.7° could be ascribed to the (100) and (002) crystal planes of the graphite-like structure of CN, respectively (JCPDS 87-1526) [21]. The significant peak centered at 27.7° was related to the interplanar staking in the g-C_3_N_4_ and the minor diffraction peak at 12.6° corresponded to the in-plane tri-s-triazine unit in the g-C_3_N_4_ [22,23,24]. After the hydrothermal process, the typical XRD diffraction peak at 27.7° for as-synthesized samples remained not notably different, maintaining the strong crystallinity. However, with increasing hydrothermal time and temperature, a new peak arising around 10.6° was increased along with the gradual decrease of the peak at 12.6°. First, the peak shift from 12.6 to 10.6° corresponded to the extension of the in-plane staking structure (the melem unit) from 0.68 nm to 0.83 nm [25]. Second, the sharp peak at 10.6° could be interpreted as the growth of the extended melem unit during the hydrothermal treatment. Therefore, the extension of the melem unit started at CN-180-6 and the extended melem unit was continuously grown by the further hydrolysis process. Meanwhile, the chemical structure of the as-prepared samples was monitored by FTIR spectroscopy (Figure 3B). The intense peak at 810 cm^−1^ represented the characteristic vibration mode of the triazine unit. The absorption bands located at 1200–1600 cm^−1^ were assigned to the typical stretching mode of the C-N aromatic structure. Broad bands at 3000–3600 cm^−1^ were the result of the uncondensed terminal amino groups or hydroxyl groups [26,27]. The main peaks of the CN-180-x and CN-200-6 samples were consistent with the pristine g-C_3_N_4_, confirming that the main chemical structure of graphitic carbon nitride was maintained even after the hydrothermal treatment for a long time and at high temperatures.

Additionally, the XPS measurement was employed to further probe the chemical bonding states of all the photocatalysts. In Figure 4, the XPS data of C 1s, N 1s, and O 1s for the CN-180-x and CN-200-6 samples are shown. In the C 1s spectra, the peak centered at 287.8 eV was related to the sp^2^-bonded carbon (N–C=N), and the peak at 284.5 eV corresponded to C-C bonds [28,29]. During the hydrothermal treatment, the deconvolution of C 1s displayed other peaks centered at 289.0 eV and 286.3 eV, which were assigned to the –COOH species and the −NH_x_ groups, respectively [30,31,32,33]. The N 1s spectra of the as-prepared samples could be fitted into three characteristic peaks centered at 398.0 eV, 398.7 eV, and 400.6 eV, relating to the pyridine N (C-N=C), the bridging N atoms in N-(C)_3_, and the amino groups with H, respectively [34,35]. Meanwhile, the O 1s XPS data of the CN-180-x and CN-200-6 samples showed three prominent peaks at 530.6 eV, 531.5 eV, and 533.0 eV, which were attributed to −COOH, −OH, and the C=O species, respectively [36,37,38]. After the hydrothermal process, the intensities of the peaks at 530.6 eV and 533.0 eV were increased, indicating the addition of more O-containing functional groups in the as-synthesized samples, which was also confirmed by the EA results (Table 2). In Table 2, the rapid increase in the atomic ratio of O/N up to CN-180-9 implied the introduction of more O-functional groups onto the g-C_3_N_4_ surface. Even though CN-180-9 obtained the highest ratio of O/N compared to all the photocatalysts, its morphological structure displayed fewer defects than the CN-180-6, which was confirmed in the TEM images. These resulted in a lower S_BET_ value for CN-180-9 than that of CN-180-6 (Table 2). 

### 3.2. Optical Properties

The optical absorption properties of the as-synthesized samples were investigated by the UV–Vis diffuse reflectance spectra (UV-DRS). As shown in Figure 5A, the absorption edge of CN, CN-180-x and CN-200-6 were in the range of 300–520 nm, from the UV to the visible light region [39]. In comparison to all prepared samples, CN-180-6 displayed a red shift and broad absorption throughout the visible spectrum, corresponding to the light-harvesting improvement for the photocatalytic reaction [40]. Additionally, the energy band edges (*E_g_*) of the catalysts were estimated by the Tauc plot method and displayed in Figure 5B. The *E_g_* values for CN, CN-180-3, CN-180-6, CN-180-9, and CN-200-6 were estimated at 2.72, 3.09, 2.83, 3.21, and 3.11 eV, respectively, which slightly enlarged with the integration of the O groups during the hydrothermal treatment. Moreover, the valence bands (*E_VB_*) and the conducting band (*E_CB_*) of the CN-180-x and CN-200-6 samples were conducted according to the empirical equation as follows [18]: (2)$$E_{VB}=\chi -E^{e}+0.5E_{g}$$
(3)$$E_{VB}=E_{CB}+E_{g}$$
where χ possessed the absolute electronegativity of the semiconductor (about 4.73 eV); $E^{e}$ was the energy of free electrons (about 4.5 eV). Hence, the *E*_*VB*_ values for CN-180-3, CN-180-6, CN-180-9, and CN-200-6 were 1.78, 1.64, 1.84, and 1.79 eV vs. NHE, while the related *E*_*CB*_ were −1.32, −1.18, −1.37, and −1.33 eV vs. NHE, respectively (shown in Appendix A).

Electrochemical impedance spectroscopy (EIS) was an effective analysis to access the charge transfer efficiency of the photocatalysts. As depicted in Figure 5C, the diameter of the arc radius for the as-prepared samples gradually declined from CN, CN-180-3, and CN-180-6. The EIS Nyquist plot of CN-180-6 revealed the smallest semicircle arc radius, suggesting the highest separation and transfer efficiency of the electron–hole pairs [41,42]. Interestingly, the CN-180-9 and CN-200-6 samples exhibited higher O-containing functional groups than the pristine g-C_3_N_4_; nevertheless, the arc radius of CN-180-9 and CN-200-6 were larger than that of the bulk, which demonstrated that the growth of the extended melem unit during the hydrothermal process could directly contribute to increasing the g-C_3_N_4_ surface resistance, resulting in the retardation in the transfer of charge carriers. To confirm the photogenerated electron transfer dynamics, the time-resolved PL decay spectra were investigated and fitted by a tri-exponential function (shown in Figure 5D). The average lifetime (τ) of charge carrier of the as-prepared samples was obtained by the following Equation (4):(4)$$\tau =\frac{A_{1}\tau_{1}+A_{2}\tau_{2}+A_{3}\tau_{3}}{A_{1}+A_{2}+A_{3}}$$
where *A*_1_, *A*_2_, and *A*_3_ are the corresponding amplitude, and *τ*_1_, *τ*_2_, and *τ*_3_ are the lifetime [43]. The CN-180-6 displayed slower exponential decay kinetics and its τ value was 5.61 ns which was much higher than that for CN (5.44 ns), CN-180-3 (5.46 ns), CN-180-9 (5.15 ns), and CN-200-6 (3.73 ns). The results implied that the formation of the O-containing melem unit during the hydrolysis/oxidation reaction initially greatly prolonged the lifetime of the photocarrier, enhancing their potential for contribution to the photocatalytic reaction [26,44,45]. Nevertheless, the longer hydrothermal reaction time and higher temperature negatively reduced the τ value due to the continuous growth of the extended melem unit. It was remarkable that the longer lifetime of the photocatalyst demonstrated the reduction recombination rate of the photogenerated electron, and thus improved the photocatalytic reaction [46,47]. In addition, the longer TRPL lifetime for CN-180-6 was further well confirmed with its highest charge transfer efficiency (EIS results) and high photodegradation performance. 

### 3.3. Photodegradation Performance

Figure 6A illustrates the photocatalytic activity of the pristine g-C_3_N_4_, CN-180-x, and CN-200-6 over the degradation of TC. Firstly, the removal of TC was investigated for 60 min under the dark phase to obtain the adsorption–desorption equilibrium. The TC concentration remained unchanged without a light source. After that, the photodegradation tests were conducted under visible light irradiation (λ > 420 nm). In Figure 6B, the TC removal efficiency of the as-prepared samples was exhibited. The photocatalytic activity of the CN-180-x and CN-200-6 was enhanced at initially in the order of CN < CN-180-3 < CN-180-6 (67.61, 84.60, and 97.17%, respectively) and declined dramatically for the CN-180-9 > CN-200-6 (57.91 and 23.30%, respectively). 

The reaction kinetic behaviors of the as-prepared samples were also investigated and demonstrated in Figure 7. According to the pseudo-first-order kinetic equation, all the fitted lines had a high linearity value (R^2^ > 0.98) (Figure 7A and Appendix A), which showed that the photocatalytic activity can be well modeled by the equation. The reaction rate constant (k) values are displayed in Figure 7B. The k values of TC photodegradation for pristine CN, CN-180-3, CN-180-6, CN-180-9, and CN-200-6 were 0.0066, 0.0118, 0.0236, 0.0058, and 0.0023 min^−1^, respectively. The k value for degradation of TC over CN-180-6 was about 3.7 times higher than that of CN. It was elucidated that the introduction of O-functional groups onto the melem unit during the hydrothermal process initially increased the reaction rate of the photodegradation, resulting in the highest photocatalytic efficiency of CN-180-6 (97.17%). However, the growth of the extended melem unit despite the abundant addition of O-containing functional groups suppressed the TC photodegradation over CN-180-9 and CN-200-6, which was directly correlated to the charge transfer efficiency. As displayed in Figure 8, the recyclability and stability of the CN-180-6 were evaluated for four continuous cycles (600 min). Under visible light, the photodegradation rate of CN-180-6 was firmly unchanged, demonstrating high activity and stability, making it a promising photocatalyst for TC degradation.

To investigate the contribution of different reactive species involved in the photocatalytic activity, triethanolamine (TEOA), isopropyl alcohol (IPA), and p-benzoquinone (BQ) were utilized to quench h+, •OH, and •O_2_^−^, respectively [48,49,50]. As illustrated in Figure 9, under visible light irradiation, the introduction of IPA and TEOA slightly inhibited the TC removal efficiency from 97.17 to 80.91 and 84.18%, respectively, revealing that the •OH and h+ played a minor role in the TC photodegradation. However, a notable decrease in TC removal (15.86%) was obtained with the addition of BQ, suggesting that •O_2_^−^ was the dominant species in the photocatalytic process. The electron in the CB of CN-180-6 could be excited from the VB and then reduced the O_2_ to form the •O_2_^−^ radicals based on the more negative CB potential (−1.18 eV vs. NHE) compared to the standard redox potential of O_2_/•O_2_^−^ (−0.33 eV vs. NHE), which was in good agreement with the band structure (Appendix A) [51,52]. 

## 4. Conclusions

In this work, we modified g-C_3_N_4_ via the hydrothermal treatment under different conditions and applied them for the photodegradation of TC. We found that the hydrothermal treatment induced not only the addition of sufficient O-containing groups into g-C_3_N_4_ but also the formation of extended melem units on g-C_3_N_4_. The latter inhibited the charge transfer on g-C_3_N_4_, resulting in a high recombination rate of photogenerated electron–hole pairs. Moreover, the porous crack structure of the extended melem units optimized the specific surface area, which enhanced the visible light absorption and the TC adsorption ability in the photocatalytic degradation. The best efficiency of CN-180-6 for the TC degradation under visible light irradiation was achieved at 97.17% because CN-180-6 contained more abundant O-functional groups and less extended melem unit on g-C_3_N_4_ with the highest specific surface area. Furthermore, from the reactive species test, •O_2_^−^ was proven as a main reactive species involving in the TC degradation.

## Figures and Tables

**Figure 1 nanomaterials-12-02945-f001:**
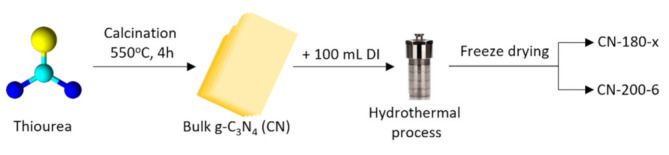
The synthesis of CN and photocatalysts.

**Figure 2 nanomaterials-12-02945-f002:**
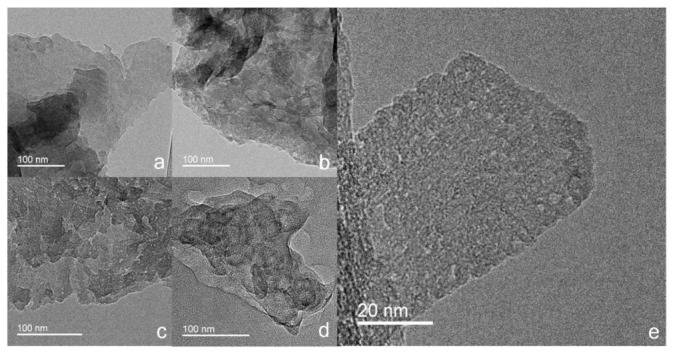
TEM images of CN-180-3 (**a**), CN-180-6 (**b**,**e**), CN-180-9 (**c**), and CN-200-6 (**d**).

**Figure 3 nanomaterials-12-02945-f003:**
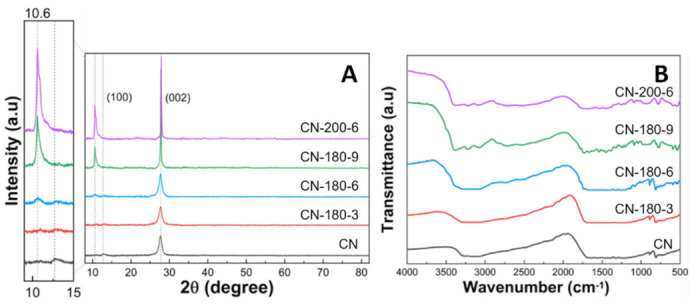
(**A**) XRD patterns and (**B**) FT-IR spectra of CN-180-x and CN-200-6 samples.

**Figure 4 nanomaterials-12-02945-f004:**
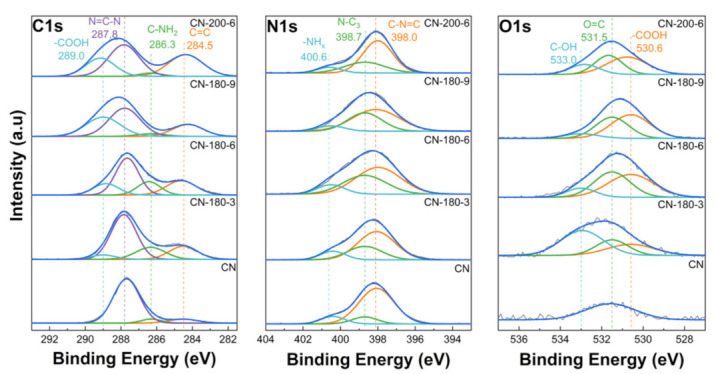
XPS data of C 1s, N 1s, and O 1s for all the photocatalysts.

**Figure 5 nanomaterials-12-02945-f005:**
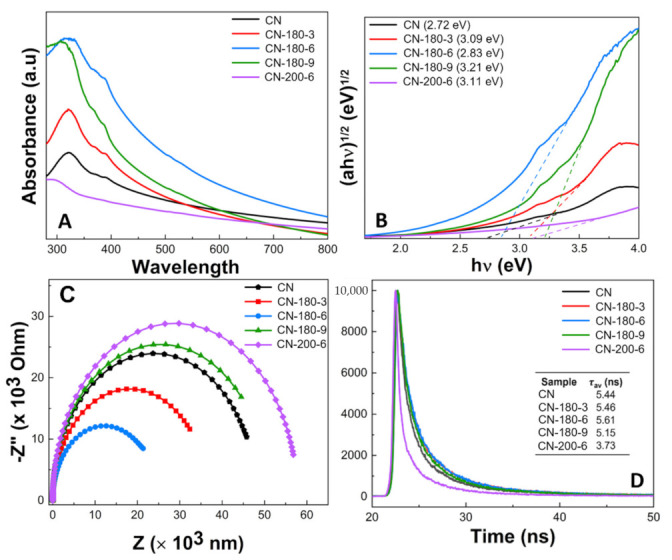
(**A**) UV–Vis DRS absorption spectra, (**B**) Tauc plots for the band gap, (**C**) the EIS Nyquist plots, and (**D**) time-resolved fluorescence decay spectra in the ns time scale with excitation 400 nm (inset table: the calculated average fluorescence lifetime (τ_av_)) of as-synthesized CN-180-x and CN-200-6 samples.

**Figure 6 nanomaterials-12-02945-f006:**
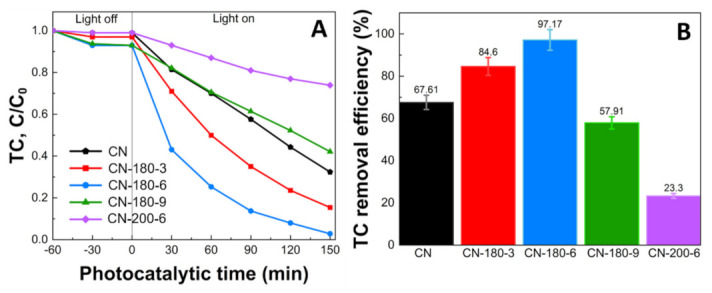
Photodegradation efficiency of TC by CN-180-x and CN-200-6 under visible light irradiation (140W xenon lamp, λ > 420nm) (**A**) C/C_0_ vs. t (Initial condition: 20 mg L^−1^ TC, 0.2 g L^−1^ catalyst, pH = 4.40, 20 °C), and (**B**) TC removal efficiency (%).

**Figure 7 nanomaterials-12-02945-f007:**
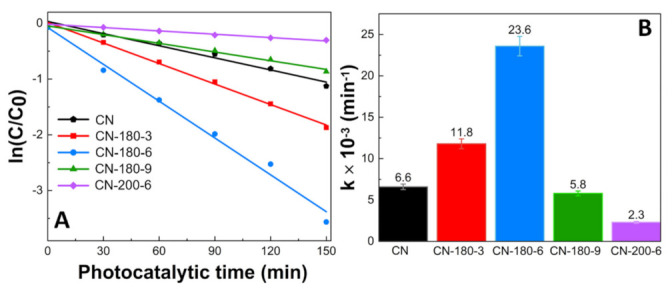
(**A**) The pseudo-first-order reaction kinetics and (**B**) rate constants (k) of as-prepared photocatalysts for TC photodegradation.

**Figure 8 nanomaterials-12-02945-f008:**
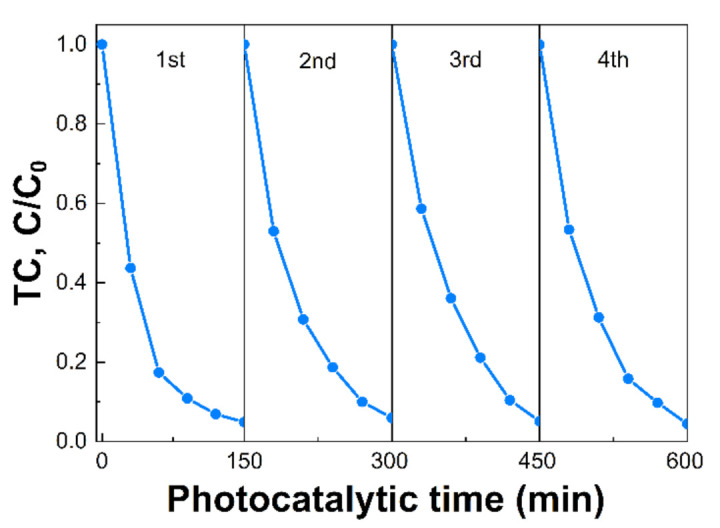
Recycling ability evaluation of CN-180-6.

**Figure 9 nanomaterials-12-02945-f009:**
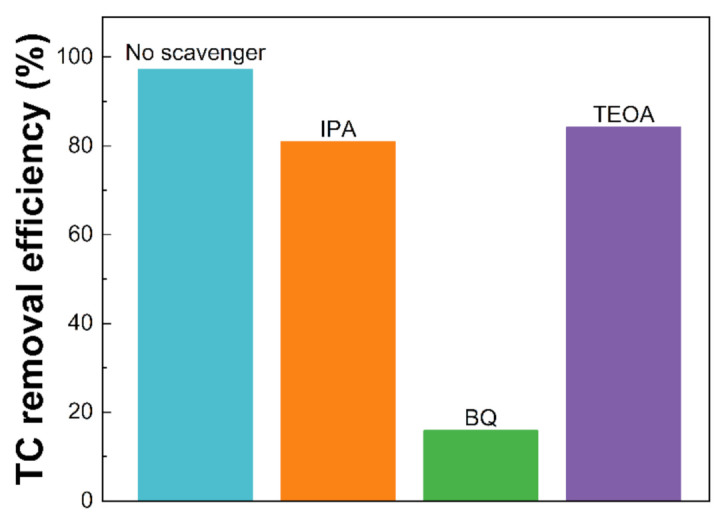
The influence of reactive species on TC removal efficiency (%).

**Table 1 nanomaterials-12-02945-t001:** Specific surface area, pore volume, and average pore size of photocatalysts were determined via N_2_ adsorption-desorption isotherm measurements.

Samples	S_BET_ (m^2^/g)	V (cm^3^/g)	L (nm)
CN	28.5	0.20	0.030
CN-180-3	59.0	0.35	0.012
CN-180-6	75.0	0.60	0.005
CN-180-9	27.5	0.11	0.002
CN-200-6	5.03	0.06	0.002

**Table 2 nanomaterials-12-02945-t002:** C/N and O/N atomic ratios of all prepared samples were obtained from elemental analysis.

Samples	Element Analysis				
	Atom (wt%)			Atomic Ratio	
	C	N	O	C/N	O/N
CN	33.8	60.3	4.5	0.65	0.07
CN-180-3	32.3	58.2	7.6	0.65	0.11
CN-180-6	29.7	53.3	13.5	0.65	0.22
CN-180-9	28.5	50.4	17.6	0.66	0.31
CN-200-6	31.5	56.1	10.3	0.66	0.16

## Data Availability

The data presented in this study are available on request from the corresponding author.

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
