# Peer review of "The Growth of Extended Melem Units on g-C3N4 by Hydrothermal Treatment and Its Effect on Photocatalytic Activity of g-C3N4 for Photodegradation of Tetracycline Hydrochloride under Visible Light Irradiation"

_nanomaterials, 2022, doi:10.3390/nano12172945_

Round 1

Reviewer 1 Report

g-C3N4 is a popular photocatalytic material, and modification of g-C3N4 by various methods and elements is the center that many people are working on. In this paper, melem units was added to g-C3N4 by hydrothermal method, and its photodegradation activity was tested, which is worthy of publication.

However, there are some problems:

1. Some abbreviations are used arbitrarily. What does AT mean? Then, using the same numbers (3,6,9,200) as suffix to represent different dimensions (time and temperature) is easy to cause confusing.

2. The characterization part appears to be too long in one paragraph.

3. How does the activity in this manuscript comparing to similar catalysts?

4. What is the purpose that uses  200 oC as a reference temperature? What will happen if the temperature is decreased to 150oC?

Reviewer 2 Report

Review of the Manuscript ID nanomaterials-1877557

1. Abstract

“Herein, we report the growth of extended tri-s-triazine units (melem units) on g-C3N4 by hydrothermal treatment and its effect on the photodegradation efficiency of tetracycline hydrochloride (TC). The AT-x samples prepared in different hydrolysis times and temperatures are characterized by multiple physicochemical techniques and their photodegradation performance was evaluated under visible light irradiation.”

Should be corrected as follows:

“In this work, the growth of extended tri-s-triazine units (melem units) on g-C3N4 by hydrothermal treatment and its effect on the photodegradation efficiency of tetracycline hydrochloride (TC) is investigated. The AT-x samples prepared using different hydrolysis times and temperatures, and they are characterized by multiple physicochemical techniques, as well as their photodegradation performance was evaluated under visible light irradiation.”

2. Conclusions

“Extended tri-s-triazine units were successfully grown on g-C3N4 by hydrothermal treatment. We successfully induced

The most advantageous hydrothermal conditions for the formation of g-C3N4 melem units were established, resulting in the best photocatalytic performance in photodegradation of tetracycline hydrochloride (TC).”

Should be substituted with:

“We successfully induced the most advantageous hydrothermal conditions for the formation of g-C3N4 melem units, resulting in the best photocatalytic performance.”

3. Please correct the references according to journal requirements. Some references are missing all the authors, some references have articles number, not pages, etc.

Reviewer 3 Report

The research represents good work on the modification of graphite-like carbon nitride to increase photocatalytic activity in the oxidation of tetracycline. The work is logically structured, the samples are characterized by modern methods, and the kinteic plots are reliably approximated. In the process of studying the manuscript, I had only a number of minor remarks.

1. In the abstract there are undeciphered abbreviations. In general, it would be better to write the abstract more "concentrated", with a clear and concise description of the achievements of the authors.

2. Table 1. Too many significant digits in BET values.

3. For work on photocatalytic oxidation, the question always arises of the mineralization of the pollutant. It would be nice to do at least one experiment to analyze total organic carbon (TOC).

4. Experiments with a standard oxidation photocatalyst, TiO2 Evonik P25, should be conducted.

Reviewer 4 Report

In this work, the author reported the extended growth of melem units during the hydrothermal reactions and their effect on the photodegradation of  tetracycline hydrochloride.  This work is well carried out and potential for publication in Nanomaterials. However, the authors are advised to modify the article with missing information as per the comments below.

1. A general information on TC photodegradation and the state of the art materials known for it should be presented in a table.

2. A scheme for the synthesis should be included.

3. It is not very clear why AT-6 performed well over others. I see there is a very slight difference on the surface area and O contents. May be structural defects or particle size should be focused.

4. A general mechanistic cycle should be included in the manuscript.

5. I suggest the author to carry out the BQ test with all the AT samples and see in which case the the photodegradation is the best. 

6. Light intensity on the sample cell should be calculated and provided.

Thank you

Round 2

Reviewer 4 Report

This revised article with synthesis scheme looks great. It would be nice if the author consider my other comments and revise the article. However, I understand in doing additional experiments and drafting out new information is not easy sometimes particularly with all sort of barriers. In this conditions, I am happy to accept this work as it is.

Thanks